# Wet Compression Study for an Aero-Thermodynamic Performance Analysis of a Centrifugal Compressor at Design and Off-Design Points

**Hyun-Su Kang** **, Sung-Yeon Kim and Youn-Jea Kim ***

School of Mechanical Engineering, Sungkyunkwan University, Suwon 16419, Korea;
hskang1504@skku.edu (H.-S.K.); ksy203@skku.edu (S.-Y.K.)
* Correspondence: yjkim@skku.edu

**Abstract:** In this study, to analyze the effect of wet compression technology on the aero-thermodynamic performance of a centrifugal compressor, a numerical analysis study was conducted on the design point and off-design point. Wet compression technology sprays water droplets at the inlet of the compressor. During the compression process, water droplets evaporate, reducing the heat of compression and reducing the compression work, which improves the efficiency of the compressor. In wet compression technology, detailed research is needed for the application to compressors because the droplet behavior affects the internal flow. The main parameters for wet compression technology are the droplet size and injection rate selection, and the flow inside the compressor changed by the droplet behavior was analyzed. When the droplet size and injection rate were changed at the design point and the off-design point, it was confirmed that a small droplet size was effective in both areas, and it was confirmed that the performance improved as the flow rate increased. The internal flow changed greatly depending on the size of the droplet. As a result, the centrifugal compressor to which the wet compression technology was applied had a lower outlet temperature than dry compression at both the design point and the off-design point and had increases in the pressure ratio and efficiency. However, the surge margin decreased by about 2% in the surge region. The reason is that due to high-speed rotation, particles move in the outer diameter direction and are driven into a tip-leakage flow, and many stagnant flows occur without flowing into the main flow. Through the study results, it was possible to understand the effects of wet compression technology on the performance and efficiency increase of centrifugal compressors and the effects of particle behavior on the internal flow of the compressor at the off-design point.

**Keywords:** centrifugal compressor; wet compression; flow angle; two-phase flow; droplet evaporation

## 1. Introduction

Compressors are used in a variety of industries. It is a very important component, especially in the energy field. Generally, compressors consume up to 40% of the energy consumed by the entire energy system or power plant. For this reason, lots of research has been done to improve compressor performance.

Mainly, research on improving the performance of the compressor has been conducted, with a lot of research on improving the pressure ratio or efficiency by redesigning the impeller or diffuser, which is a key element. However, most redesign methods have many engineering costs because aerodynamic performance and structural and vibration issues must be considered.

Wet compression technology is a way to compensate for these shortcomings. It does not require a separate component redesign and has proven to be effective just by injecting water droplets at the inlet. Especially in the power generation system, when the external temperature rises by 1 degree, the power of the power plant is reduced by up to 1%. The external environment cannot be controlled, but if water is injected to lower the

compressor inlet temperature, the temperature of the compressor internal flow channel can be lowered, and a decrease in the temperature inside the flow channel has the effect of reducing compression work. Methods to lower the compressor inlet temperature include intercooling, evaporative cooling, inlet fog cooling, and wet compression. Among these methods, the most effective method is wet compression [1], where un-evaporated droplets flow into the compressor and evaporate and cool by the high-temperature, high-pressure environment of the internal flow path.

The wet compression technology was first proposed by Kleinschmidit [2] in 1940 and has been studied by many researchers. After that, in 1989, the wet compression technology was firstly installed in the gas turbine for the first time [3]. Jolly [4] experimentally applied wet compression technology to a gas turbine, resulting in a 9% power improvement. Bettocchi et al. [5] compared the performance curves of wet compression and conventional air compression and experimentally proved that the pressure ratio increased by 2% when the wet compression method was used. Neupert et al. [6] visualized the flow of droplets floating inside the compressor using a particle analyzer (Laser doppler anemometry, LDA, or Particle doppler anemometry, PDA). Recently, with the development of numerical analysis technology, it has become possible to study the interior of a compressor using wet compression technology in three dimensions. Furthermore, Sun et al. [7] advanced the numerical analysis of the physical interaction between droplets and blades.

Wet compression technology has been studied to have various advantages when applied to a gas turbine's axial compressor. However, research on applying wet compression technology to centrifugal compressors is relatively insufficient. In fact, there are more than 1000 cases of wet compression applied to a gas turbine [8], but it is still difficult to find cases applied to centrifugal compressors. Since a centrifugal compressor has different flow characteristics from axial compressors, additional studies are needed to apply wet compression. If wet compression technology is applied to the centrifugal compressor, wet compression technology can be effectively applied due to the advantage of the centrifugal compressor, which can be used in various operating conditions.

Since late 2010, several researchers have been conducting research to apply wet compression technology to a centrifugal compressor. Surendran et al. [9] applied a wet compression technique to a low-speed centrifugal compressor using numerical analysis and showed that the compression work decreased when the droplet inject amount was 3% or more. Sun et al. [10] numerically studied the effect of supercritical wet compression on the performance of centrifugal compressors used in compressed air energy storage systems.

Although the wet compression technique has been proven effective by various studies, it has problems, such as droplet collision and a reduction in the operating area [11]. In addition, the compression process changes due to droplets inside the compressor, and the flow angle changes. If the flow angle at the outlet of the impeller is largely changed, abnormal phenomena, such as a stall and surge, may occur along with the deterioration of the compressor performance, making it difficult to operate the compressor [12].

Wet compression technology has been proven to be effective by many studies, but in-depth research on droplet variables is required to apply wet compression due to complex compressor internal flow phenomena, such as droplet behavior. Therefore, in this study, a wet compression simulation was performed for not only the design point, but also the off-design point, and the internal thermodynamic performance analysis, as well as the internal flow angle and internal flow distribution, were considered. In addition, as a detailed goal, the variables in wet compression technology, diameter and injection rate, were calculated for the design point and surge area, and the effect of the variables on the flow and performance of the centrifugal compressor was studied.

## 2. Methodology

### 2.1. Governing Equations

#### 2.1.1. Continuous Phase

In this paper, a numerical analysis was conducted to analyze the wet compression effect. For the numerical analysis, ANSYS CFX 2020R2, a commercial program, was used to solve the Reynolds-Averaged Navier–Stokes (RANS) equations. The RANS equation consists of a continuity equation, momentum equation, and an energy equation. The governing equations that describe the motion of continuous fluid are as follows:

Continuity equation:

$$\frac{\partial \rho}{\partial t} + \nabla \times (\rho \vec{v}) = S_M \tag{1}$$

where $\rho$ is the fluid density, and $\vec{v}$ is the velocity field of the fluid. The right term ($S_M$) represents the source term of mass transfer from the dispersed phase to the continuous phase caused by the evaporation of droplets.

Momentum equation:

$$\frac{\partial (\rho \vec{v})}{\partial t} + [\nabla \times (\rho \vec{v}) \vec{v}] = -\nabla p + \nabla \times \vec{\tau} + S_F \tag{2}$$

where p is the pressure, and $S_F$ represents the external body force. The external body force represents the momentum transfer between the dispersed phase and continuous phase. Furthermore, $\vec{\tau}$ means shear stress and is expressed as follows:

$$\vec{\tau} = \mu[(\nabla \vec{v} + \nabla \vec{v}^t) - \frac{2}{3} \nabla \times \vec{v} I] \tag{3}$$

Energy equation:

$$\frac{\partial (\rho e)}{\partial t} + \nabla \times (\rho \vec{v} e) = \nabla(\lambda \nabla t) + \rho \nabla \times \vec{v} + \nabla \times (\vec{v} \times \tau) + S_E \tag{4}$$

where $\lambda$ is the thermal conductivity, and $S_E$ is the energy source term, which represents the energy exchange between the dispersed phase and the continuous phase.

#### 2.1.2. Dispersed Phase

The numerical analysis of wet compression is a two-phase flow, in which two different fluids flow, and is calculated using the Eulerian–Lagrangian method for calculating droplet information along with the RANS equations. Three equations are needed to exchange information on the motion and evaporation of droplets. The first is an equation to describe the motion of droplets, a heat transfer equation to describe the heat transfer effect of continuous and dispersed phases, and, finally, the mass transfer equation to calculate the interphase mass transfer.

- Particle transport equation

The particle transport equation is used to elucidate the motion of the droplet. The forces exerted on the droplet in the equation are forces by gravity, the centrifugal and Coriolis forces, the virtual mass force, and the pressure gradient force. However, in this study, only the forces by gravity and the centrifugal and Coriolis forces, which have a pronounced influence on the droplet movement, were used due to the high rotational speed of the fluid machine. The particle transport equation is written as follows:

$$m_d \frac{du_d}{dt} = F_D + F_R \tag{5}$$

where $m_d$ is the droplet mass, and $u_d$ is the droplet velocity. $F_D$ and $F_R$ are the forces by gravity and the centrifugal and Coriolis forces mentioned above and are expressed as follows:

$$F_D = \frac{1}{2}C_D\rho_{air}d^2|u - u_d|(u - u_d) \tag{6}$$

$$F_R = m_d[-2\Omega \times u_d - \Omega \times (\Omega \times r)] \tag{7}$$

where d is the droplet diameter, u is the fluid velocity of continuous phase, $\Omega$ is the angular velocity, r is the position vector, and $C_D$ is the drag coefficient, which is determined by the following empirical correlation [13]:

$$C_D = \frac{24\left(1 + 0.15Re_{d_r}^{0.687}\right)}{Re_{d_r}}, \ Re \ \leq 1000 \tag{8}$$

- Heat transport equation

The heat transfer between the continuous phase and dispersed phase is calculated from the following equation:

$$m_p C_d \frac{dT_d}{dt} = \pi d\lambda Nu(T - T_d) + h_l\frac{dm_p}{dt} \tag{9}$$

where $C_d$ is the specific heat of water droplets, $T_d$ is the temperature of droplet, and $h_l$ is the latent heat of vaporization. Nu is the Nusselt number, which is the ratio of convective to conductive heat transfer. In this study, Nu was used through the Ranz–Marshall correlation, given by [14]:

$$Nu = 2 + 0.6Re^{0.5}\left(\mu\frac{C_P}{\lambda}\right)^{\frac{1}{3}} \tag{10}$$

- Mass transport equation

The mass transport equation calculates the mass change that occurs in the process of phase change of droplets. In this study, two mass transfer correlations, according to the boiling point, were used. Mass transfer is decided by boiling point. The boiling point is determined through an Antoine equation given below [15]:

$$\log_{10} p_{sat} = A - \frac{B}{T + C - 273.15} \tag{11}$$

where $p_{sat}$ is the saturation pressure, and A–C are the constant coefficients.

When the droplet temperature is above the boiling point, the mass transfer rate can be written as follows:

$$\frac{dm_d}{dt} = -\frac{\pi d\lambda Nu\left(T - T_p\right)}{h_{fg}} \tag{12}$$

Conversely, the mass transfer rate below the boiling point is as follows:

$$\frac{dm_d}{dt} = \pi d\rho_v DSh\frac{M_v}{M} \log\left(\frac{1 - f_p}{1 - f}\right) \tag{13}$$

where $\rho_v$ is the vapor density, and D is the diffusivity coefficient. Sh is the Sherwood number, and M and $M_v$ are the air and water vapor molar mass, respectively. f and $f_p$ are the mole fraction of the air–water mixture and the mole fraction of the water droplets, respectively.

### 2.2. Reference Model

The NASA CC3 model was used as an industrial air compressor to validate the effect of wet compression on the centrifugal compressor, and its performance and efficiency have been proven through various experiments [16,17]. The model impeller consists of 15 blades

and 15 splitters, and the diffuser consists of 24 vanes. The design point of the analysis model is a 4.17 pressure ratio and 21,789 rpm, and the operating flow rate is 4.54 kg/s. Detailed dimensions are shown in Table 1 and Figure 1.

**Table 1.** Specification of the geometrical parameters.

| Impeller | |
| --- | --- |
| Number of impeller blades | 15 |
| Number of splitter blades | 15 |
| Impeller LE radius at hub ($r_{Hub}$) | 41 mm |
| Impeller LE radius at shroud ($r_{Shroud}$) | 105 mm |
| Impeller TE radius ($r_1$) | 215 mm |
| **Diffuser** | |
| Number of diffuser blades | 24 |
| Diffuser LE radius ($r_2$) | 232 mm |
| Diffuser TE radius ($r_3$) | 363 mm |
| Blade height ($D_H$) | 17 mm |

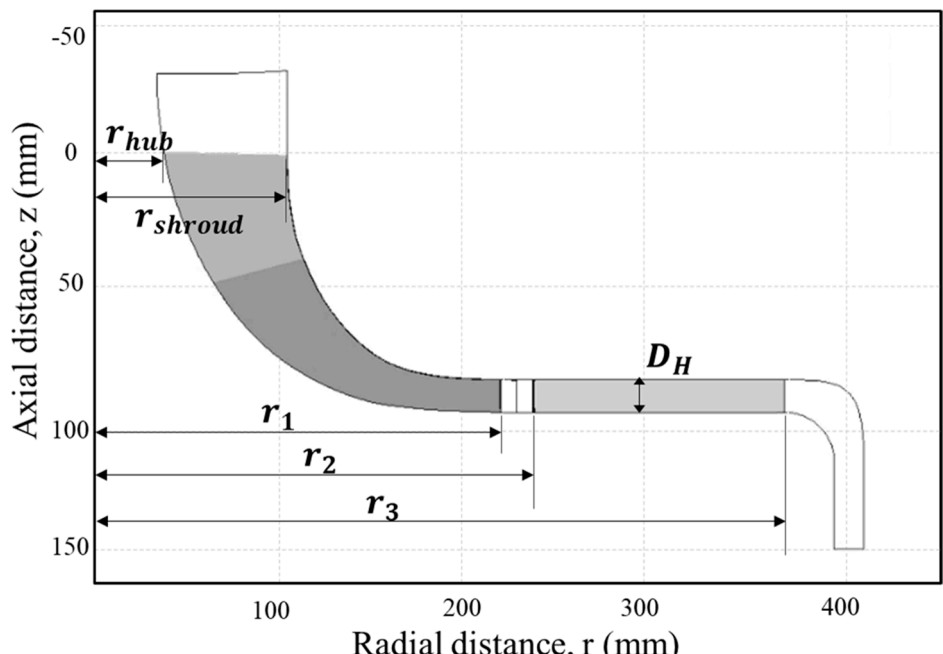

**Figure 1.** Specification of the geometrical dimensions.

*2.3. Grid System*

Securing a reliable grid system is one of the important processes in the CFD. In this study, structured grids for CFD were generated using ANSYS TurboGrid. Figure 2 shows the grid of the compressor, including the impeller and diffuser. For the blade flow path, an O-grid system was used for the surface near blades, and an H-grid system was used for other regions. In addition, a grid dependency test was conducted to minimize the influence of the grid (see Figure 3). The pressure ratio and isentropic efficiency were chosen as the evaluation index, and it was confirmed that the result value converged at about $1.2 \times 10^6$ elements. Therefore, a final 557,798 grids were used in the impeller region and 509,082 grids in the diffuser region, and a total of $1.2 \times 10^6$ grids were used.

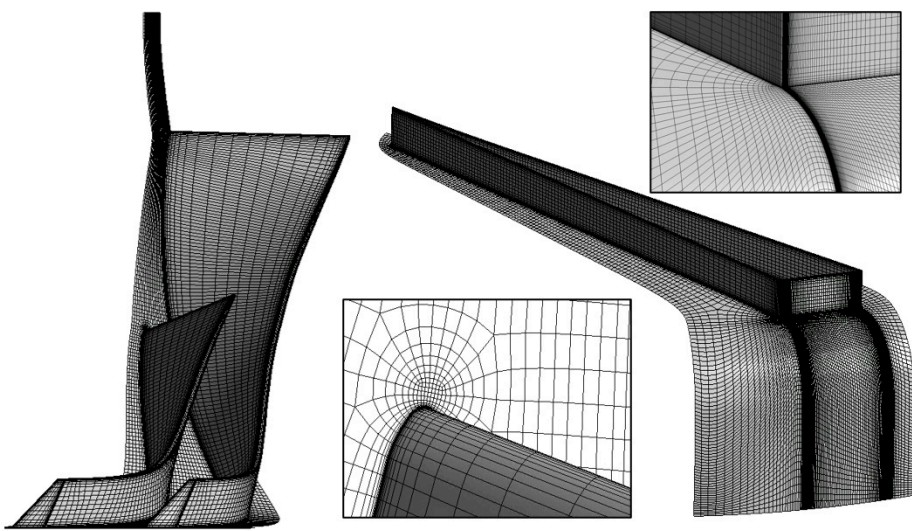

**Figure 2.** Grid systems of the numerical analysis.

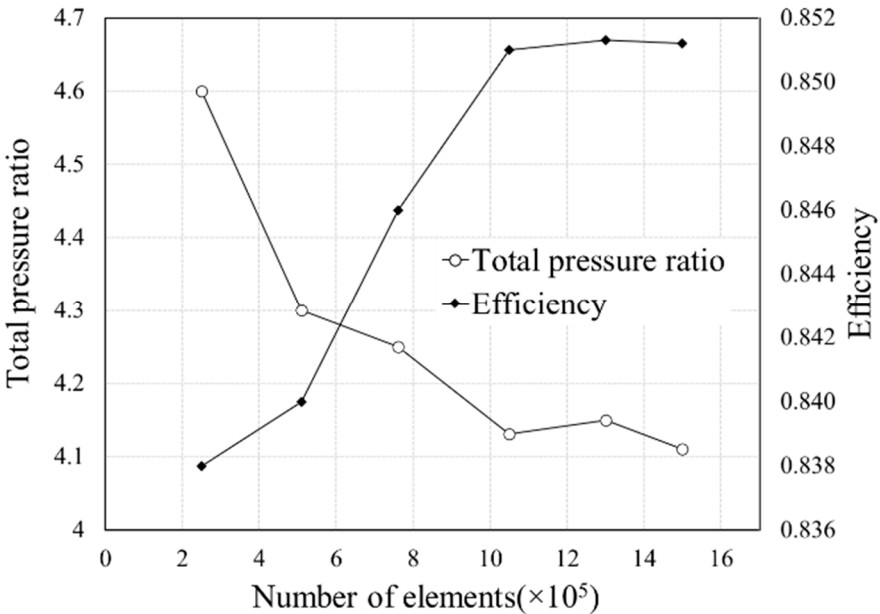

**Figure 3.** Performance results of the grid dependency test.

*2.4. Boundary Conditions*

In this study, a steady state analysis was performed to verify the performance curve. The MRF (moving reference frame) technique was used to analyze the rotation of the compressor. The SST (shear stress transport) model was used as the turbulence condition. Wet compression technology is a two-phase flow, in which droplets are injected with compressed air. For this reason, boundary conditions for fluids and droplets injection are required. The continuous phase fluid was calculated using the Eulerian method, and the injected droplets were calculated using the Lagrangian method. The calculation method through the Eulerian–Lagrangian method was calculated in a two-way coupled method, and the droplet information, such as temperature and trajectory, were calculated along with the flow. Detailed boundary conditions are described in Table 2.

**Table 2.** Boundary conditions of the centrifugal compressor.

| Continuous Phase | | |
|---|---|---|
| Rotating speed | | 21,789 [rpm] |
| Fluid | | Air (ideal gas) |
| Inlet | Total pressure | 101.325 [kPa] |
| | Temperature | 288.15 [K] |
| Outlet | Average static pressure | 410 [kPa] |
| Interface | | Stage (mixing-plane) |
| Convergence criteria (RMS) | | $1.0 \times 10^{-4}$ |
| Dispersed Phase | | |
| Droplet | | Water |
| Droplet temperature | | 288.15 [K] |
| Number of droplets | | 10,000 |
| Injection type | | Full cone |

Wet compression technology exposes droplets to gas flow inside the compressor. Therefore, breakup occurs due to the difference in velocity between gas and liquid phases. In this study, the cascade atomization and breakup (CAB) model was used to consider the slip velocity. The CAB model was developed in the Taylor Analogy Breakup (TAB) model, in which the breakup of droplets is described as a system composed of a spring-mass system and damped resonance and is known to be suitable for liquid spray analysis. In addition, the Sommerfeld collision model [16], a probabilistic collision model, was used to consider the impact caused by collisions between droplets and blades.

### 3. Validation of Aero Performance and the Two-Phase Flow Model

*3.1. Performance Curve Mapping*

To validate the numerical analysis method, the CFD results were compared with existing published data [16–18]. For comparison, the pressure ratio and isentropic efficiency, which are the compressor performance index, were compared. The pressure ratio and isentropic efficiency are defined as follows:

$$\text{Pressure ratio(PR)} = \frac{p_{out}}{p_{inlet}} \tag{14}$$

$$\text{Isentropic efficiency(Eff)} = \frac{W_c}{W_{i,c}} \tag{15}$$

The comparisons of the CFD and experimental results are shown in Figure 4 and showed a good agreement.

*3.2. Two-Phase Flow Model Validation*

In this chapter, a two-phase flow model also needed to be verified. To validate the two-phase flow model, the NACA 0012 model was chosen. The experimental result [19] and CFD results were compared when NACA 0012 was rained on (refer to Figure 5). As a result of validation, it shows that an angle of attack (AOA) was performed on the airfoil from −5° to 9°, and all relative errors came within 7%. It was judged that the two-phase flow model used in this study is acceptable for this study.

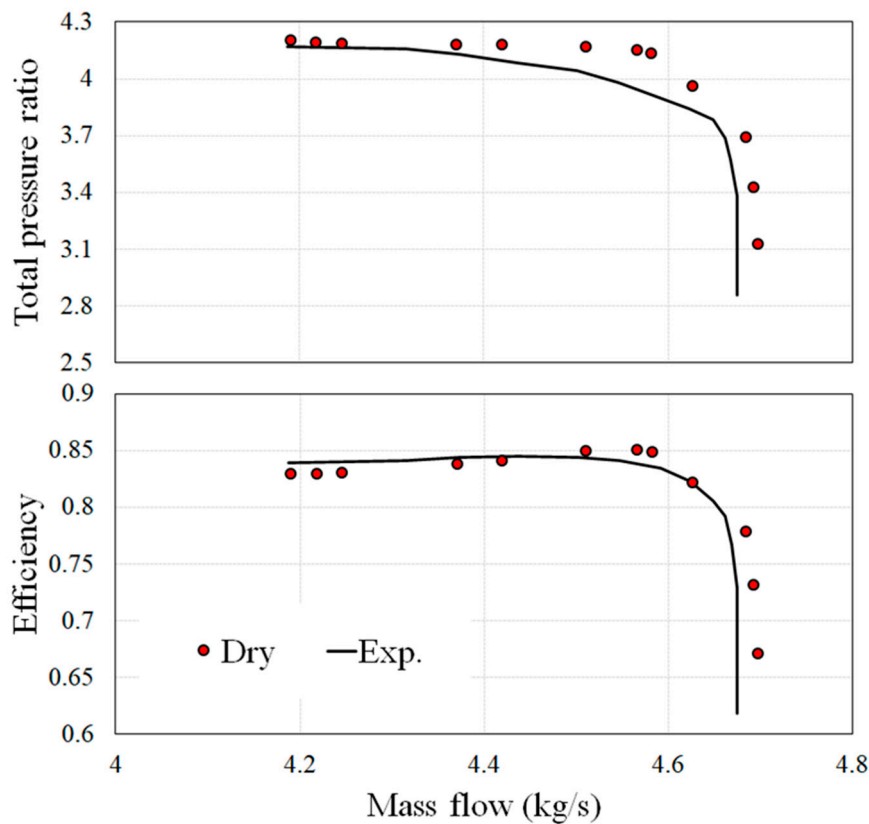

**Figure 4.** Comparison results of the total pressure ratio and efficiency between the CFD and experimental results.

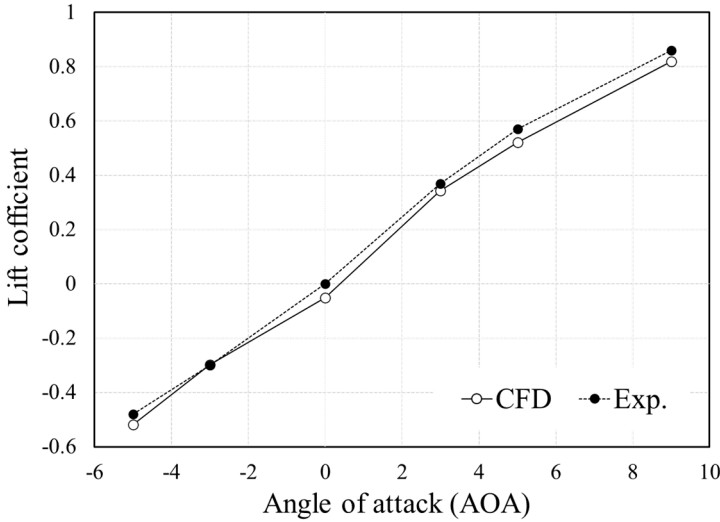

**Figure 5.** Comparison results of the lift coefficient curve for the NACA 0012 airfoil for rain conditions between the experiment [19] and CFD.

## 4. Results of Wet Compression at Design Point

*4.1. Overall Performance Analysis*

In this section, the influence of wet compression technology on aerodynamic performance at the design point was analyzed. To compare the effects of dry compression and wet compression, the wet compression technology injected droplets having a uniform size of 2 μm at 0.2% of the air flow rate. The actual droplet distribution consisted of various droplet sizes. The distribution of these droplets used a theoretical formula derived from

an empirical formula. In general, the numerical analysis used the Rosin–Rammler (RR) distribution [20], which has few errors and is easy to apply. Accordingly, numerical analysis was performed by applying various droplet distributions along the RR distribution, not the uniform droplet size. The RR distribution equation is expressed as follows:

$$R = \exp\left[-\left(\frac{d}{d_e}\right)^{\gamma}\right] \qquad (16)$$

where $d_e$ is the RR mean diameter, and $\gamma$ is the RR parameter. A general value of $\gamma$ for the wet compression used is 1.5~3.0, but in this study, a value of 2 was used, referring to the literature on the existing wet compression research. Figure 6 compares the droplet trajectories when the uniform droplet size is 2 μm and when the average diameter is 2 μm according to the RR distribution. When looking at the droplet trajectory, the heterogeneous atomization condition had a droplet size of up to 3 μm, and thus, the trajectory was longer than that of the uniform droplet diameter. In terms of compression ratio, isentropic efficiency, and outlet temperature, there was no significant difference. Because the droplet size distribution in the two conditions differed, it was assumed that the differences did not have a substantial impact on performance, because the change in the droplet distribution was minor. Hence, to examine the compressor performance through the motion of a specific diameter, all analyzes that were conducted were performed using a homogeneous droplet diameter.

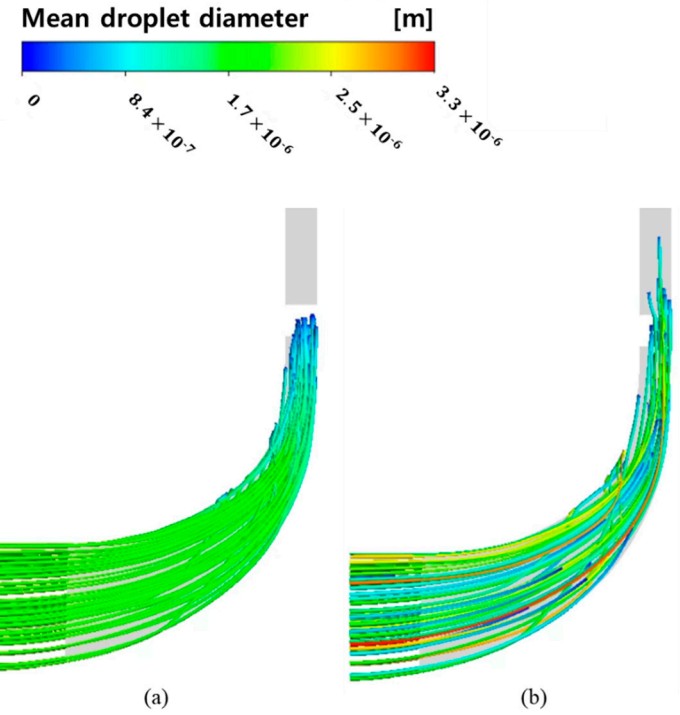

**Figure 6.** Droplet trajectories on the meridional plane under different injection conditions; (**a**) homogeneous injection and (**b**) heterogeneous injection.

The NASA CC3 model confirmed that when wet compression technology was applied to the design point, the pressure ratio and efficiency increased compared to the dry compression. At the design point, the pressure ratio improved by 0.12% compared to the existing dry compression. The mass flow rate of the air mixture increases because the temperature inside the compressor decreases by the latent heat of the droplet as the droplet is injected to increase the air density. The isentropic efficiency is an indicator of whether the temperature inside the compressor decreases. Therefore, the higher the efficiency, the clearer the heat absorption reaction induced by the latent heat of the droplet. The isentropic

efficiency increased by 1.14% from air compression. The diffuser outlet temperature was 457.71 K in air compression, and it decreased by 3.6 K to 454.12 K when wet compression was applied.

Wet compression technology affects not only the thermodynamic aspects, but also the aerodynamic aspects, due to the droplets' motion. Figure 7 shows the velocity distribution of the impeller outlet, and Figure 8 shows the velocity angle distributions at same position as shown in Figure 7.

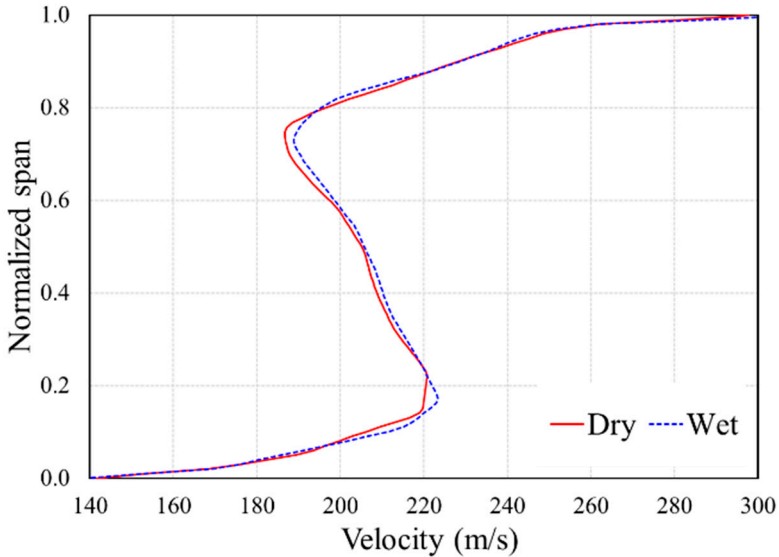

**Figure 7.** Impeller outlet velocity distribution of dry and wet compression.

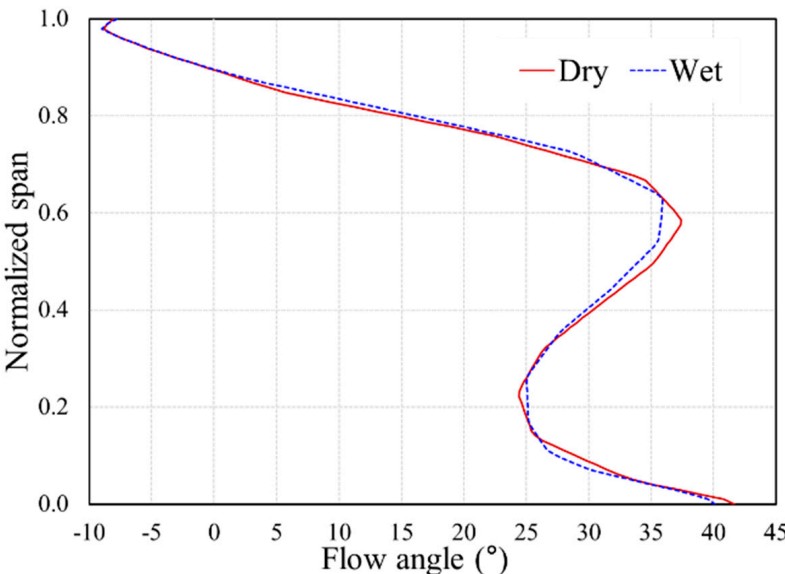

**Figure 8.** Impeller outlet flow angle distribution of dry and wet compression.

At the outlet of the impeller, the flow velocity differed by several m/s from about a range of 0.2~0.8, and overall, the speed was slower in the dry than the wet compression. Similarly, for the flow angle distribution, the flow angle difference occurred in the range of about 0.2~0.7. The presence or absence of droplets affects the flow speed and angle inside the impeller, and these changes affect not only the performance of the impeller, but also the diffuser, where kinetic energy is converted into pressure energy. From the next section, various flow rates and droplet diameters were analyzed for a detailed analysis of wet compression.

### 4.2. Effect of the Droplet Size on the Aerodynamic Performance of the Centrifugal Compressor at the Design Point

The droplet diameter is closely related to the evaporation rate and, thus, affects the performance. It is known that the appropriate droplet size in gas turbines is 5 μm to 10 μm. In addition, the smaller the droplet diameter, the less erosion is caused [21].

To scrutinize the effect of the droplet size on the centrifugal compressor performance, the flow rate, outlet temperature, pressure ratio, and isentropic efficiency were compared, varying the droplet diameter from 1~10 μm. The results were normalized for the convenience of comparison according to the droplet diameter and are shown in Figure 9. The isentropic efficiency had the highest values at the droplet size of 2 μm, and the values of other droplet diameters were normalized based on 2 μm. The outlet temperature was normalized with 6 μm and was calculated by the following equation:

$$\text{Normalized parameters} = \frac{\text{Value}}{\text{Value}_{max}} \tag{17}$$

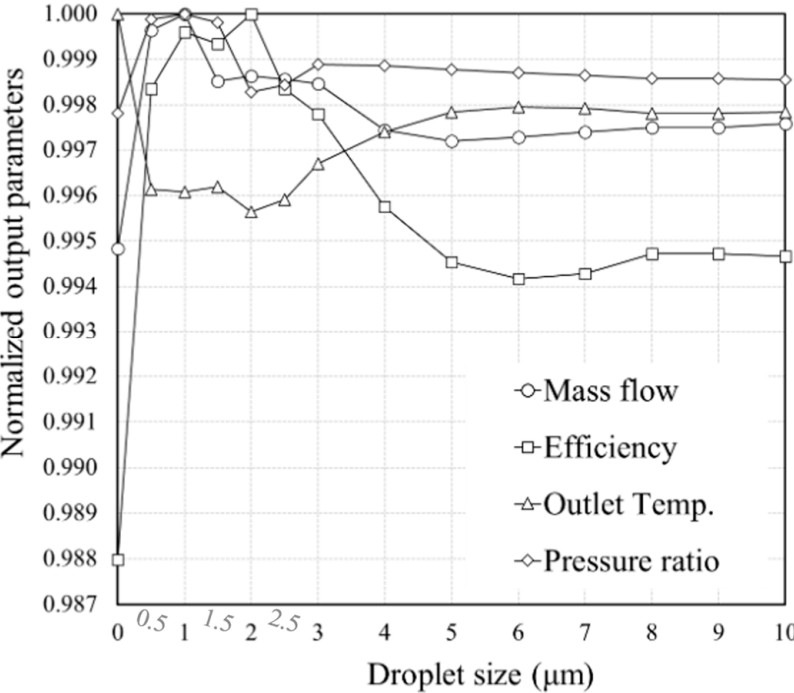

**Figure 9.** Effect of the different droplet sizes on the centrifugal compressor performance (injecting mass flow rate of 0.2%) at the design point.

To increase precision in the analysis results, the intermediate values of particle sizes having high efficiency intervals (1~3 μm) were additionally analyzed. When the droplet diameter was changed, the best aerodynamic performance was shown at 2 μm. On the contrary, the aerodynamic performance was lowest at 6 μm. The results of the droplet trajectory at 2 μm and 6 μm are compared in Figure 10.

In the case of 2 μm, all droplets were evaporated at the impeller outlet, and the particle diameter at the impeller outlet also decreased to 1 μm or less. On the other hand, when it was 6 μm, some droplets were delivered to the diffuser region without being evaporated inside the impeller. The droplet diameter at the impeller outlet showed a distribution of up to 5 μm or more. Additionally, if the diameter of the impeller is large, it is separated from the main flow upstream of the impeller, and it could be confirmed that a lot of it joins as a tip leakage flow.

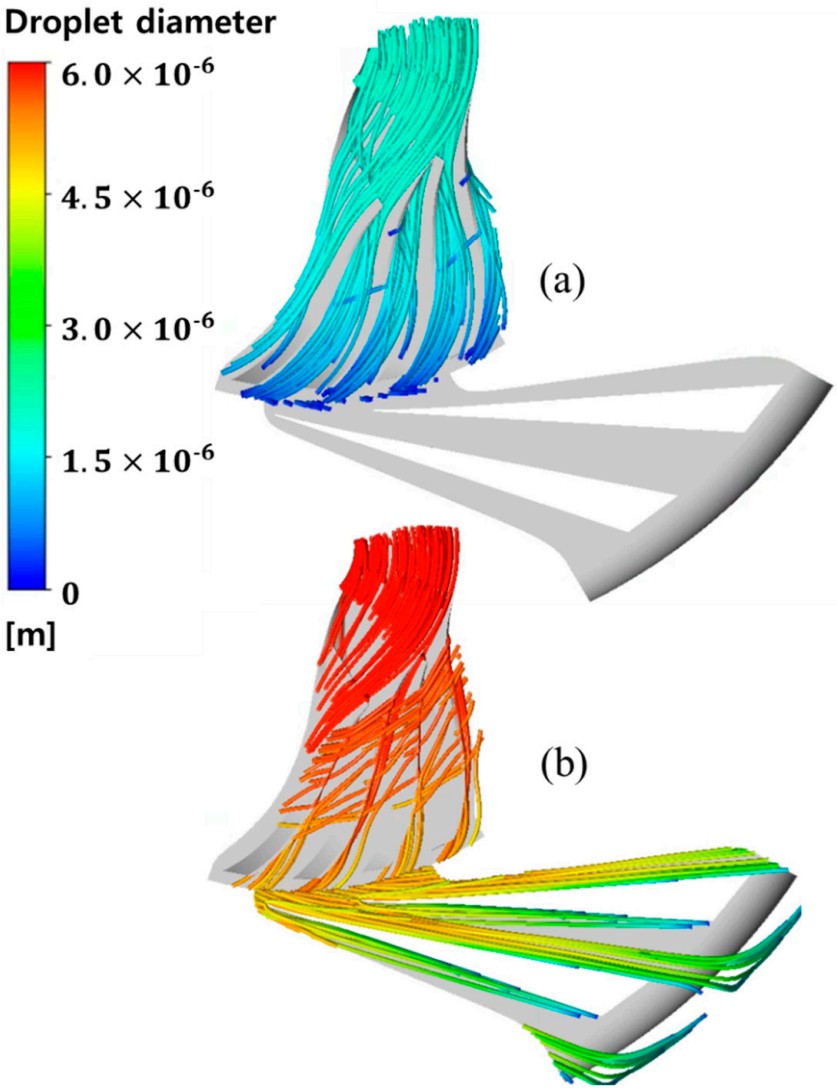

**Figure 10.** Comparison results of the droplet trajectories for the case of droplet diameter; (**a**) $d_r$ = 2 μm and (**b**) $d_r$ = 6 μm.

The droplets inside the centrifugal compressor are subjected to centrifugal force in the radial direction. The larger the droplet diameter, the more the centrifugal force is affected; hence, droplets flock toward the shroud. These results tend to be similar to previous studies on wet compression [22,23].

As the droplet diameter increases, the particle response time increases, which is greatly influenced by the centrifugal force. The particle response time refers to the time required for the particles to react to changes in fluid flow. Thus, droplets with a large diameter are not affected by the gas flow field; consequently, they deviate faster from the main flow. Figure 11 shows the mass fraction distribution of droplets with sizes of 2 μm and 6 μm in the flow direction. It was confirmed that a model with a droplet size of 6 μm was concentrated toward the gap by a large centrifugal force. Most of the droplets moved toward the tip clearance, resulting in more leakage flow, forming an unstable internal flow field. This was judged to be the reason that the performance was lower when the particle diameter was large. In wet compression technology, the larger the particle diameter, the particle response time increases to be longer, which leads to tip leakage. It was confirmed that the performance of the compressor was better when the particle diameter was small.

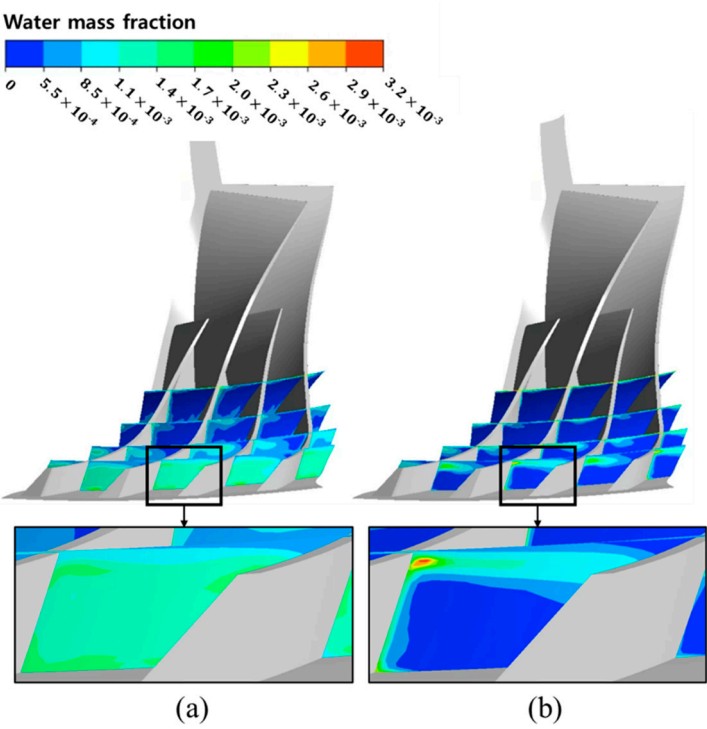

**Figure 11.** Comparison results of the mass fraction for the case of droplet diameter; (**a**) $d_r$ = 2 μm and (**b**) $d_r$ = 6 μm.

*4.3. Effect of the Mass Flow of Water Injection on the Aerodynamic Performance of a Centrifugal Compressor at the Design Operating Condition*

In this section, the calculation results, according to the injection rate, are discussed. The droplet diameter was fixed to 2 μm, having the highest efficiency at the design point, and the droplet flow rate was performed in the order of 0.2%, 0.4%, 0.6%, 0.8%, and 1.0% of the air flow rate. The increase in the amount of water injected indicates that the size of latent heat contained in water increases; hence, the cooling effect linearly improves performance. Figure 12 shows the results, and the internal temperature decreased as the flow rate increased. Figure 13 shows the temperature distribution by injection rate. The cooling effect clearly occurred in the diffuser region due to the evaporated latent heat. Accordingly, the outlet average temperature difference between 1% injection and 0.2% injection was 9.3 K. In addition, as the injection rate increases, the trajectory of the droplets inside the compressor increases. As the injection rate increases, more droplets enter the impeller region; thus, it is thought that the trajectory increases because relatively unevaporated droplets increase. Comparing the time and distance over which the actual droplet remains, the traveling time of the longest droplet was 0.004461 s when the flow rate was 1% and the traveling distance was 0.3883 m, and when it was 0.2%, it was 0.003275 s and 0.3746 m. From the design point, when water was injected at an injection rate of 0.2% of the air, most of the droplets evaporated at the end of the impeller region and in front of the diffuser region. As the injection rate increases, droplets exit to the diffuser region, but the traveling distance of the droplet is not different because most of the droplets with the longest traveling distance rotate in the centrifugal direction rather than the streamwise direction. The advantages of flow rate in wet compression are obvious, but if the amount exceeds a certain amount, a stall or flow instability might occur, so flow control is required, considering the operating schedule of the compressor [23].

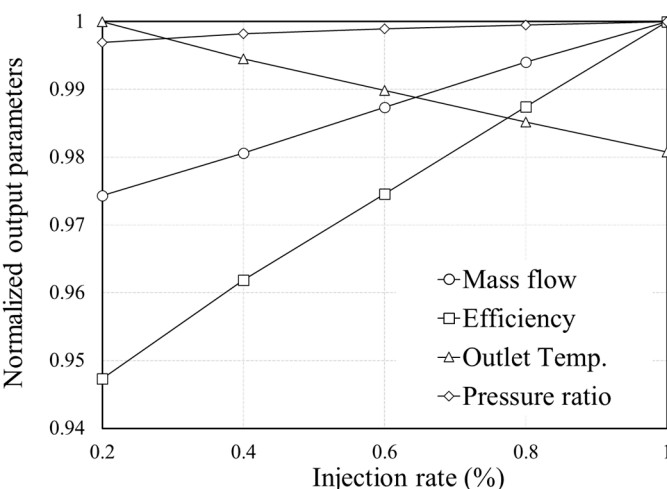

**Figure 12.** Effect of the different mass flow rates on the centrifugal compressor performance (droplet diameter, $d_r$ = 2 μm) at the design point.

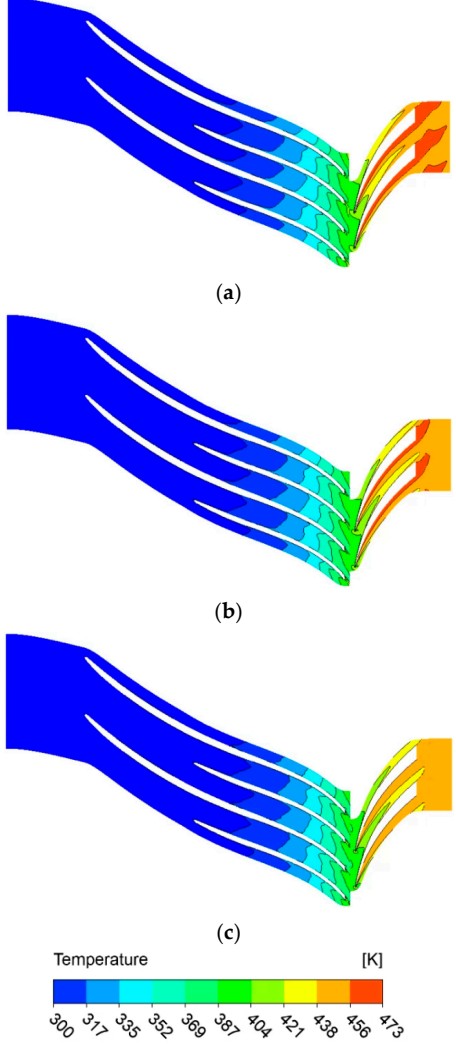

**Figure 13.** Comparison results of the temperature contour under different droplet mass flow rates at a span 50%. (**a**) 0.2% (0.00092 kg/s); (**b**) 0.6% (0.00276 kg/s); (**c**) 1% (0.0046 kg/s).

## 5. Results of Wet Compression at the Off-Design Region

*5.1. Performance Analysis at the Off-Design Operating Condition*

In this section, the effect of wet compression technology on a wide range of flow rates, from design flow rates to flow rates near surge, was analyzed. Figure 14 illustrates the dry and wet compression performance curves for the pressure ratio and efficiency. In the wet compression technology, based on the previous section, the size of the droplet was 2 μm, and the injection rate was 0.2% of the air flow rate. In wet compression, both the pressure ratio and the isentropic efficiency moved to the right, which means that more flow rates had passed compared to the same performance. When wet compression technology is applied, the surge condition moves toward a high flow rate. To confirm this, the stall margin was evaluated, which is expressed as follows:

$$\text{SM} = \left[ 1 - \frac{\text{PR}_{\text{peak}}}{\text{PR}_{\text{stall}}} \times \frac{m_{\text{stall}}}{m_{\text{peak}}} \right] \times 100\% \tag{18}$$

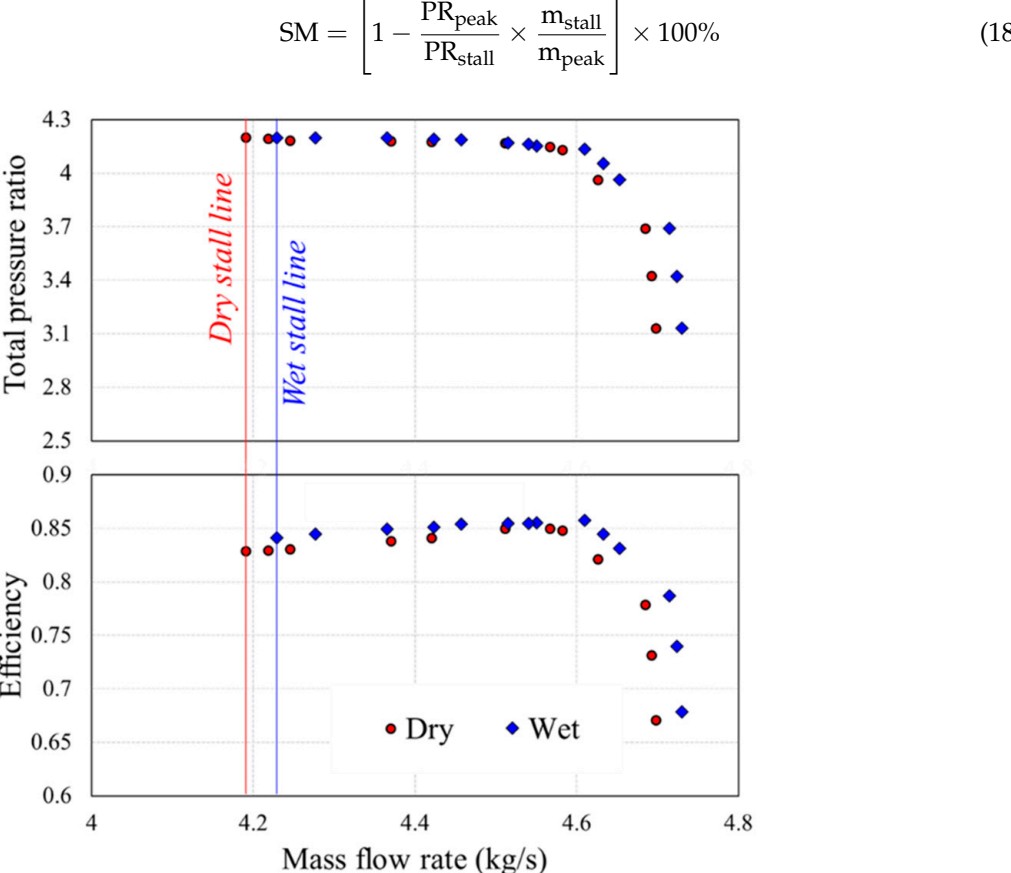

**Figure 14.** Comparison results of the performance curve between the dry and wet compression.

The SM value decreased by 2% in the wet compression compared to the dry compression. In general, the surge phenomenon occurs at a low flow rate point and occurs in the impeller and diffuser regions, causing an instability in the flow. Accordingly, the droplet motion changes the flow inside the impeller to reduce SM. Figure 15 shows the leakage flow occurring at the top of the tip clearance (red arrow) near the surge point. In the case of (a) dry compression, a few parts of the leakage flow generated from the top of the shroud went to the adjacent blade, but in the case of (b) wet compression, it can be noted that a large amount of the leakage flow generated from the top moved to the adjacent blade and formed a blockage (red circle). As a result, in wet compression, there is relatively more leakage flow that does not flow downstream and is stagnant.

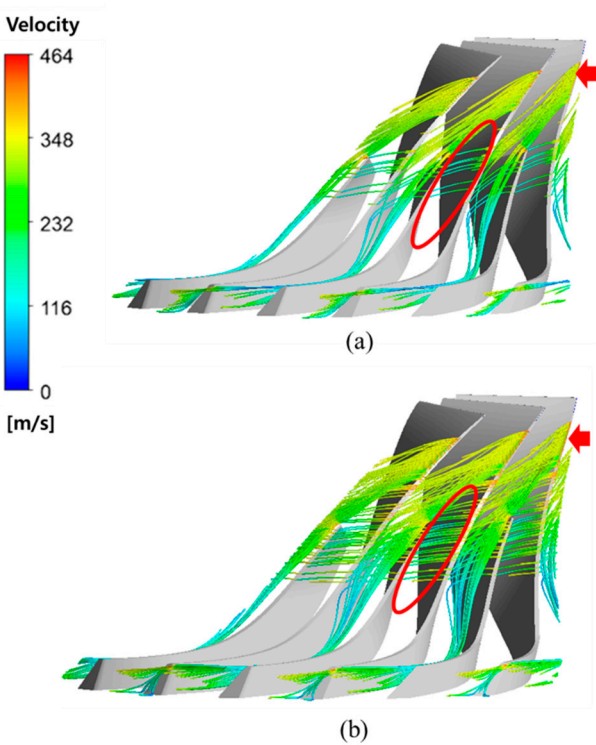

**Figure 15.** Comparison results of the leakage flow between the (**a**) dry and (**b**) wet compression.

Figures 16 and 17 show the eddy viscosity in the dry and wet compression at a span of 0.9 (normalized between the hub to shroud). Figure 16 shows that a relatively higher eddy viscosity occurred in (b) the wet compression compared to (a) the dry compression near the outlet of the impeller.

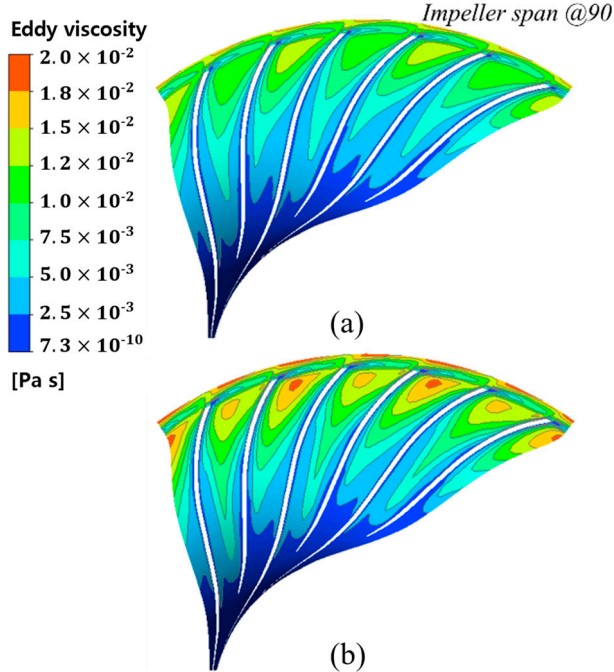

**Figure 16.** Eddy viscosity distribution between the (**a**) dry and (**b**) wet compression at the stall region.

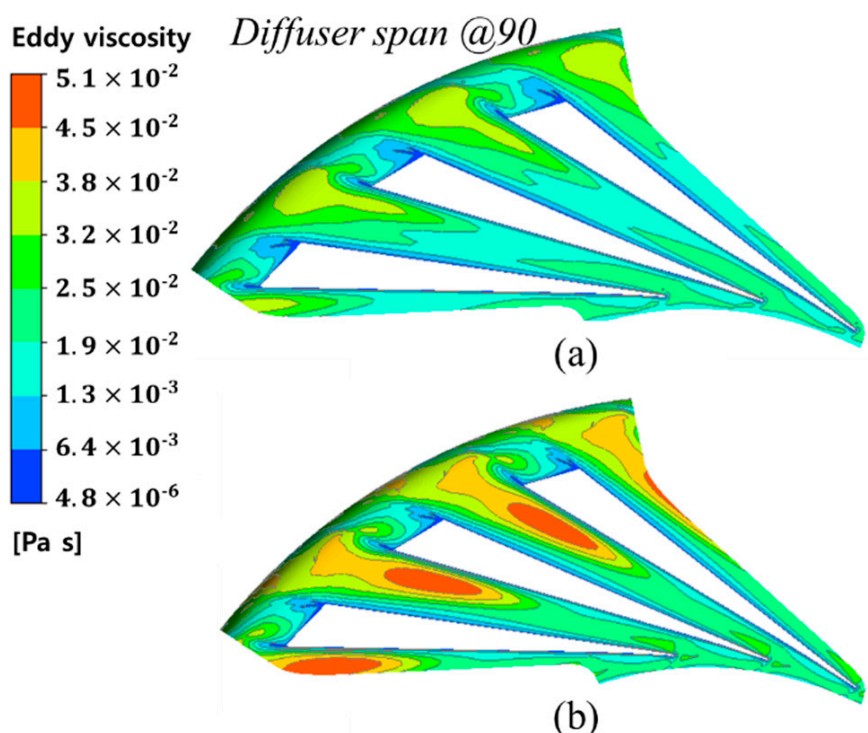

**Figure 17.** Diffuser eddy viscosity distribution between the (**a**) dry and (**b**) wet compression at the stall region.

Similarly, a higher eddy viscosity occurred near the exit of the diffuser in the wet compression, shown in Figure 17b. An unstable vortex field generated from the upstream was accumulated, and more unstable flow occurred in the wet compression. Blockages and strong vortices in the compressor internal flow path cause a numerical stall.

In the wet compression, the leakage flow developed from the upstream did not flow downstream as well as that of the dry compression, resulting in larger blockage near the span of 0.9 and a relatively larger eddy viscosity near the impeller and diffuser outlets. For this reason, the stall margin was lower in the wet compression than that in the dry compression.

### 5.2. Performance Analysis at the Off-Design Operating Condition

Figure 18 shows the performance according to the size of the droplet to analyze the performance according to the size of the droplet at the off-design point. As with the design point, the size of the droplet was changed from 1 to 10 μm to confirm the flow rate, outlet temperature, pressure ratio, and isentropic efficiency. Unlike the design point, it showed the best performance at 1 μm in the surge area. Results also showed that the performance decreases as the droplet diameter increases. When the droplet diameter grows, the number of droplets that have not evaporated inside the centrifugal compressor influences performance, just as it does with the design point. Figure 19 shows the average volume fraction in the direction of the streamwise, and the droplet diameter with 1 μm evaporated at the streamwise location of 0.6. On the other hand, when the droplet diameter was 10 μm, droplets existed at the end of the diffuser. As the droplet diameter increases, the point of evaporation is delayed. The latent heat size of the diffuser region cannot be used for cooling when the droplets leave the diffuser region in an evaporated state; hence, the smaller the droplets are injected, the higher the performance.

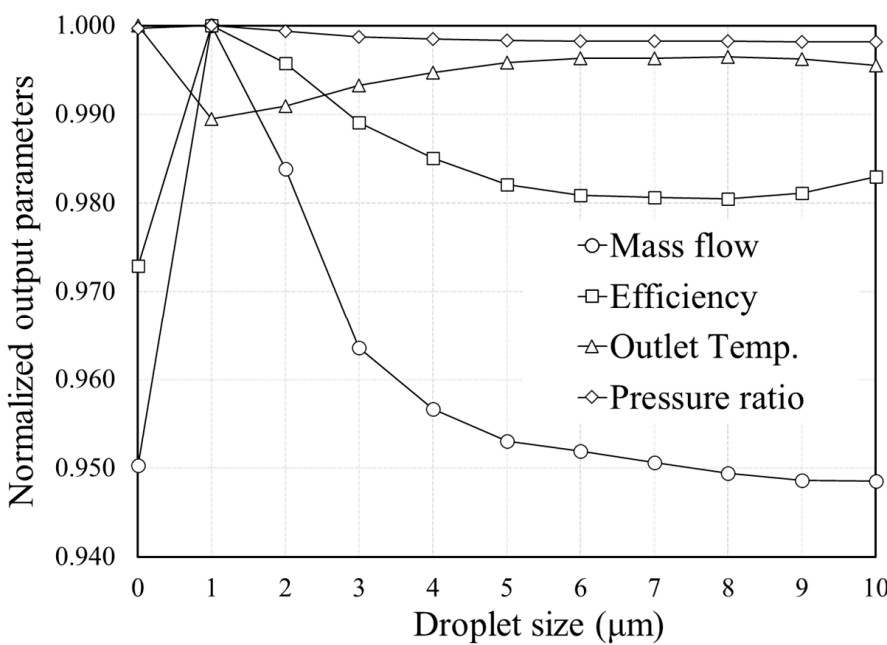

**Figure 18.** Effect of the different droplet sizes on the centrifugal compressor performance.

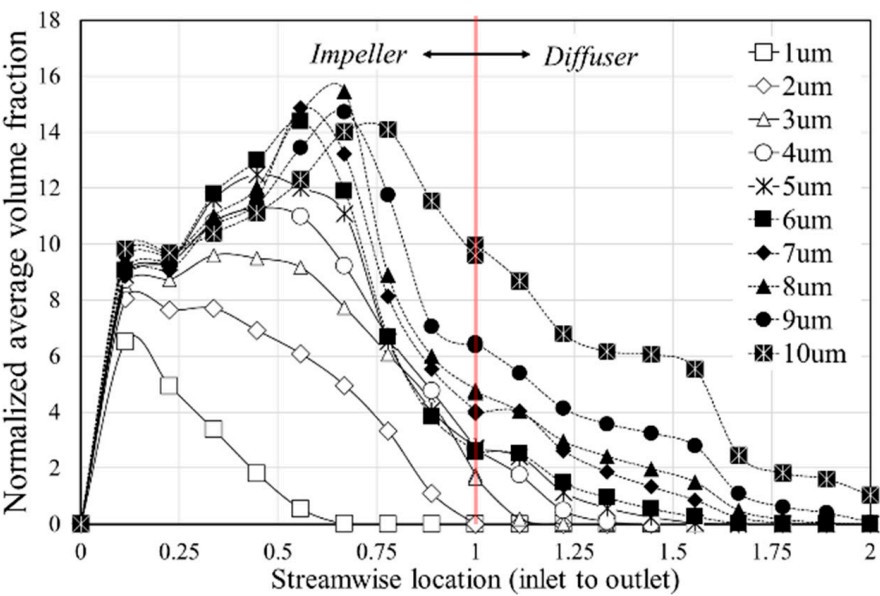

**Figure 19.** Averaged volume fraction along the streamline position with different droplet sizes.

*5.3. Performance Analysis at the Off-Design Operating Condition*

Figure 20 shows the performance change when the injection rate was changed by 0.2~1.0% in the surge area. As with the previous design point, the flow rate linearly increased. Figure 21 shows the average volume fraction for each streamwise location. As the flow rate increases, the mass fraction of water increases in the impeller region. However, as the injection rate increased, the evaporation position of all droplets did not change significantly. Increasing the flow rate increases the amount of latent heat to increase the cooling effect but does not change the evaporation position of the droplet. Consequently, the size of the droplet has the greatest influence on the evaporation of the droplet. Results also showed that the outlet temperature decreased to 458.239 K, 454.499 K, 454.865 K, 449.395 K, and 447.021 K, respectively, as the injection rate increased. Figure 22 shows the performance for the design point and the surge area to compare with the change in

the injection rate. In both the surge area and the design point, the pressure ratio, outlet temperature, and isentropic efficiency showed similar trends, but the mass flow rate of air showed a steeper difference as the water injection rate increased. This is thought to have delivered more air compressed by the cooling effect in the surge region where the internal temperature is higher.

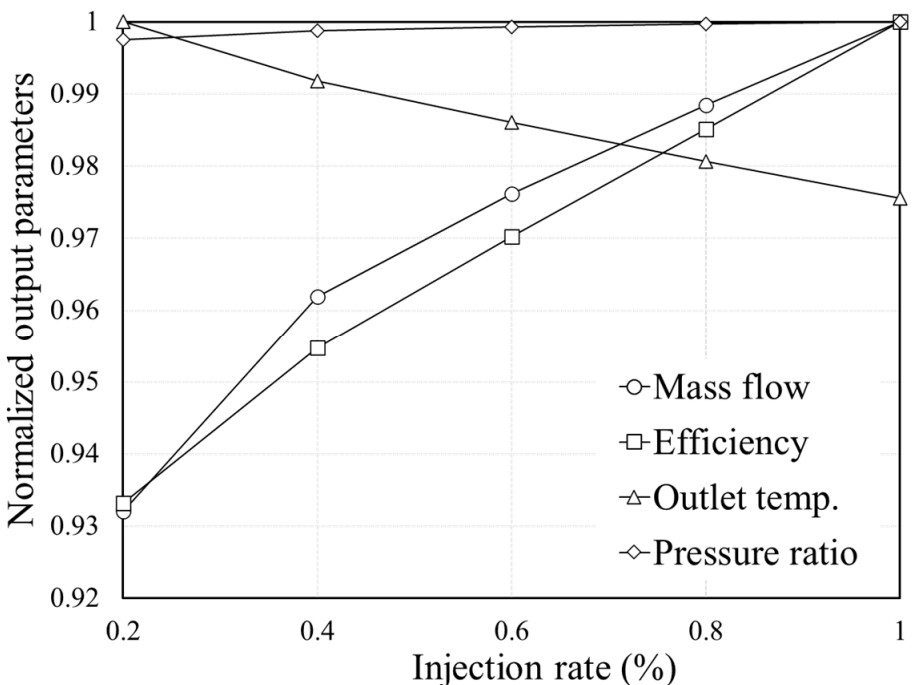

**Figure 20.** Effect of the different mass flow rates on the centrifugal compressor performance.

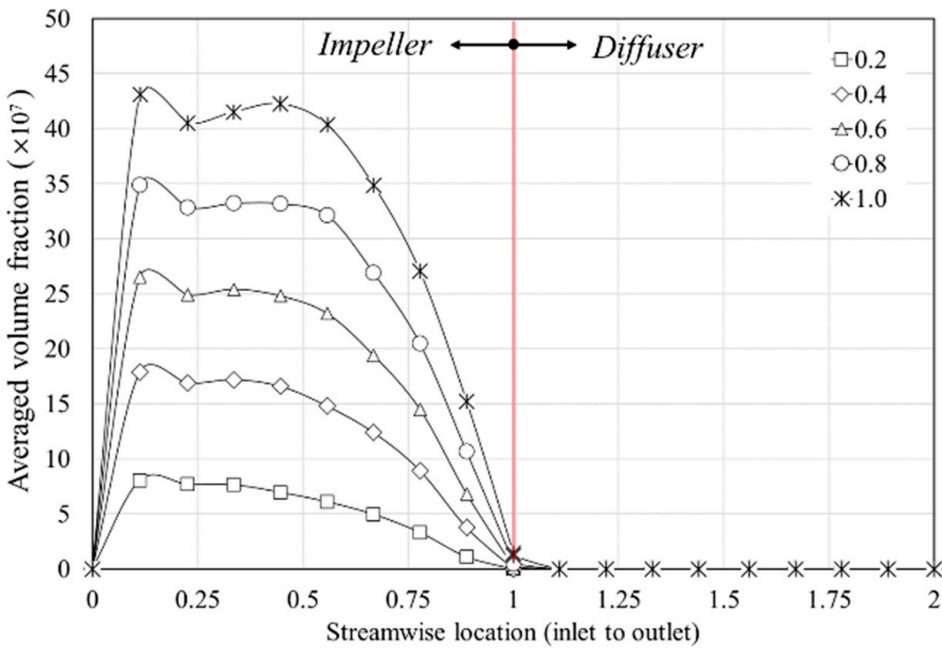

**Figure 21.** Averaged volume fraction along the streamline position with different water injection rates.

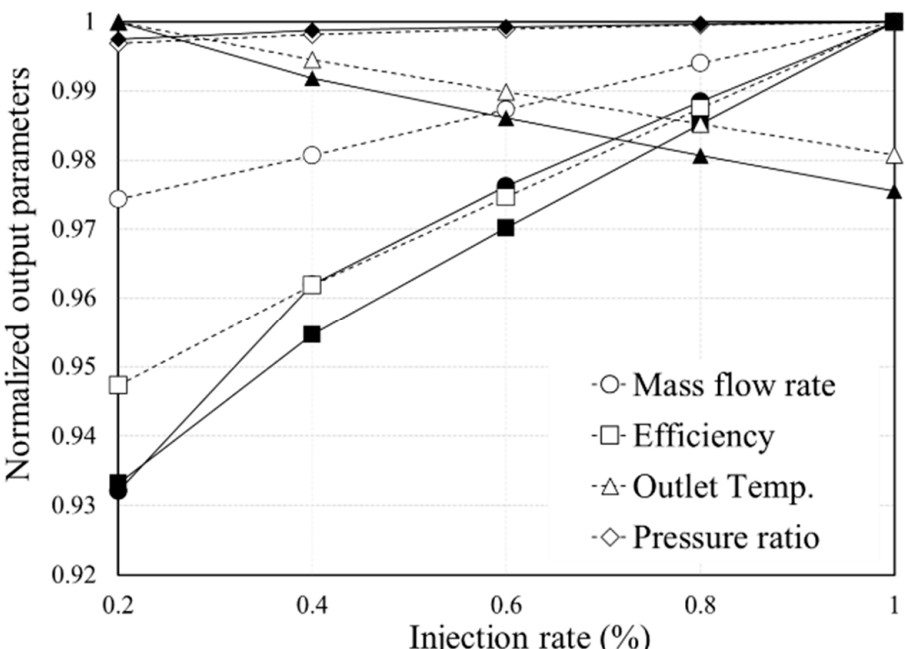

**Figure 22.** Effect of the different mass flow rate on the centrifugal compressor performance between the design point and surge region (Filled symbol: design point; open symbol: surge region).

## 6. Conclusions

In this study, the wet compression technology that improves compressor performance by injecting droplets to the inlet area was investigated. Through this, the effect on the aerodynamic performance of the centrifugal compressor was considered, and the results are summarized as follows:

(1) A droplet having a diameter 2 μm at the design point was injected into the compressor inlet area at 0.2% of the air flow rate (4.54 kg/s). At the design point, the pressure ratio increased by 0.12% and the isentropic efficiency by 1.14% compared to the air compression. The variable that has a dominant influence on the efficiency increases is the evaporation of the droplet. Droplet evaporation reduces the internal temperature of the compressor by the amount of latent heat and, consequently, increases the efficiency.

(2) At the design point, the performance was compared by changing the droplet diameter by 1 to 10 μm. The droplet diameter was best at 2 μm, and the performance deteriorated from the droplet diameter with a size of 6 μm or more. This is because the evaporation rate decreased as the surface area of the droplet was different while having the same mass flow rate. In addition, the droplet diameter with 6 μm quickly departs from the main flow due to its relatively large mass. As a result, it was established that most of the droplets moved toward the tip, generating more leakage flow and forming an unstable flow. The performance improves linearly as the water injection rate increases. Accordingly, the outlet average temperature difference between a 1% spray and 0.2% spray was 9.3 K.

(3) A numerical analysis was performed by applying wet compression technology to various operating points by changing the static pressure in the outlet area of the centrifugal compressor. Wet compression has been shown to affect not only the design point, but also the off-design point, including near the surge. In the surge region, wet compression generated more leakage flow according to droplet motion, and it reduced SM by 2% compared dry compression. The effective droplet size in the surge area was 1 μm. Like the design point, the small droplet size was effective, and as the droplet diameter increased, the unevaporated droplet increased. The water injection rate affected the performance in the surge region.

**Author Contributions:** Supervision, Y.-J.K.; writing original draft, H.-S.K. writing review and editing, H.-S.K. and S.-Y.K. All authors contributed to the manuscript. All authors have read and agreed to the published version of the manuscript.

**Funding:** This research was supported by the SungKyunKwan University and the BK21 FOUR (Graduate School Innovation) funded by the Ministry of Education (MOE, Korea) and National Research Foundation of Korea (NRF).

**Conflicts of Interest:** The authors declare no conflict of interest.

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
