# Peer review of "Wet Compression Study for an Aero-Thermodynamic Performance Analysis of a Centrifugal Compressor at Design and Off-Design Points"

_processes, doi:10.3390/pr10050936_

Round 1

Reviewer 1 Report

Wet compression is an interesting topic since it can help improve aerodynamic performance of compressors. This paper deals with a cetrifugal compressor with air and water droplets. Two phases theory and CFD are used to study flow field and performance. The whole paper is written clearly. There are still some problems in the paper. The authors should address them.

  1. A lot of information is given for the dispersed phase. However how to calculate SM, source term in Eq(1) and SE in Eq.(2), i.e. how to relate the air phase with water phase?  Is this theory available in CFX?
  2. In Fig.7 and Fig.8, are parameters invlolving velocity and flow angle relative or absolute?
  3. There are errors in Lines 114,148, 177, 191,221,245,249,252,354,355,458,475 .
  4. In Fig18, there is no unit for the abscissa.    

Author Response

Thank you

Reviewer 2 Report

Specific Comments and Questions:

  1. Section 2. Is the grid convergence study conducted also at wet conditions?
  2. Section 2. Table 2. Please, specify the convergence criteria: RMS or MAX.
  3. Section 2. Table 2. Why is considered 10,000 of droplets?
  4. Figure 9 and 18, Units of x-axis magnitude are missing.
  5. Section 4. Figure 9. What is the level of the uncertainty of these KPIs? Since the region of 0-3 micrometers of droplet size is under study, it is recommended to increase the resolution to obtain intermediate results (0.5, 1.5, 2.5 should be enough). Under the reviewer’s opinion, this is of importance since along the discussion and conclusions the droplet size of 2micrometers is considered as the one that provides max eff, when it should be 2+-0.5.
  6. Section 5. Figure 15. The visualization of this effect in this chart is poor: the comparison of both velocity (relative) streamlines are not well visualized. Please, adapt this plot or, otherwise, it may be deleted from the manuscript since the corresponding paragraph may not be aligned with this chart.

Global evaluation:

The paper is overall well written and well organized, the results are presented in a clear manner and the scientific treatment of the analysis is sound. Therefore, it is certainly recommended for publication after reviewing minor changes.

Author Response

Thank you
